# Clinical Significance and Therapeutic Challenges of *Scedosporium* spp. and *Lomentospora prolificans* Isolates in a Single-Center Cohort of Lung Transplant Recipients

**DOI:** 10.3390/jof11040291

**Published:** 2025-04-08

**Authors:** Sarela García-Masedo Fernández, Rosalía Laporta, Myriam Aguilar, Christian García Fadul, María Cabrera Pineda, Ana Alastruey-Izquierdo, Ana Royuela, Isabel Sánchez Romero, Piedad Ussetti Gil

**Affiliations:** 1Microbiology Department, Hospital Universitario Puerta de Hierro, 28222 Majadahonda, Madrid, Spain; mcabrerap@salud.madrid.org (M.C.P.); msromero@salud.madrid.org (I.S.R.); 2Pneumology Department, Hospital Universitario Puerta de Hierro, 28222 Majadahonda, Madrid, Spain; rosalia.laporta@salud.madrid.org (R.L.); myriam.aguilar@salud.madrid.org (M.A.); christian.garcia@salud.madrid.org (C.G.F.); pied2152@separ.es (P.U.G.); 3Mycology Reference Laboratory, National Centre for Microbiology, Instituto de Salud Carlos III, 28220 Majadahonda, Madrid, Spain; anaalastruey@isciii.es; 4Clinical Biostatistics Unit, Instituto de Investigación Sanitaria Puerta de Hierro-Segovia de Arana, 28222 Majadahonda, Madrid, Spain; aroyuela@idiphim.org

**Keywords:** *Scedosporium*, *Lomentospora*, transplant, lung, infection, antifungal

## Abstract

(1) Background: Emerging fungal infections associated with *Scedosporium* spp. and *Lomentospora prolificans* (S/L) are becoming more frequent and are very difficult to treat. Our objective was to analyze the frequency and management of S/L isolates in lung transplant (LTx) recipients, the patient outcomes and in vitro antifungal sensitivity. (2) Methods: We included all patients with S/L isolation during post-transplant follow-up. Data were collected from electronic medical records. All samples were cultivated on Sabouraud Chloramphenicol agar. Isolations of S/L were submitted to in vitro susceptibility tests. (3) Results: A total of 11 (2%) of the 576 LTx recipients included had at least one isolation of S/L. Classification for the 11 cases were colonization (4; 36%) and infection (7; 65%). Five infections were pulmonary (71%) and two were disseminated (29%). *S. apiospermum* complex was the most frequently occurring isolation in patients with pulmonary disease while *L. prolificans* was the most frequent in patients with disseminated disease. Ten patients were treated. The most frequent antifungal drugs used were voriconazole (n = 8) and terbinafine (n = 6). Seven patients (70%) received more than one drug. The mortality rate associated with *L. prolificans* isolation was 50% for colonization and 100% for disseminated disease. (4) Conclusions: *Scedosporium* spp. and *L. prolificans* infections are associated with high morbidity and mortality rates. New diagnostic and therapeutic tools are required to reduce the impact of these infections.

## 1. Introduction

Infectious complications are a frequent cause of morbidity and mortality in lung transplant recipients. The constant exposure of the lung graft to the external environment and the high levels of immunosuppression required to prevent acute and chronic graft rejection are the main predisposing factors to infections [1,2].

Fungal infections are less prevalent than bacterial or viral infections but have been associated with a higher mortality rate [1,3]. *Aspergillus* is the most frequently isolated fungus, but in recent years, there has been a progressive increase in infections caused by other emerging fungi, such as *Scedosporium* spp. and *Lomentospora prolificans*. The limited therapeutic options available against these emerging fungi are associated with high mortality and represent a challenge for clinicians.

*Scedosporium* spp. and *Lomentospora prolificans* are ubiquitous pathogenic fungi primarily found in soil and wastewater. These fungi have a particular ability to colonize the respiratory tract of patients with pulmonary disease and are commonly isolated in respiratory samples before and after lung transplantation. Fungal colonization is especially frequent among lung transplant candidates with cystic fibrosis. After *Aspergillus fumigatus*, *Scedosporium* spp. and *Lomentospora* are the second most frequently isolated fungal group in these patients [4,5,6].

In the post-transplant period, a significant percentage of lung transplant recipients (20–50%) show fungal colonization, with rates of invasive disease reaching up to 14% [6]. The infection develops as a result of a complex interaction between the fungus and the host’s immune system. In immunocompetent individuals, the innate immune response plays a crucial role, and infections are typically limited [7,8]. However, in patients with compromised immune systems, *Scedosporium* spp. and *Lomentospora prolificans* can produce invasive pulmonary disease and hematogenous dissemination. The ineffective immune response of the host, together with the intrinsic resistance of these fungi to most of the antifungal drugs currently available, are the main causes of the high mortality rate associated with these infections [9,10].

In the context of lung transplantation, the distinction between colonization and disease is particularly difficult. Colonization always precedes infection but is not necessarily synonymous with disease. The diagnosis of disease can be challenging due to the nonspecific nature of symptoms and the complexity of interpreting radiological findings. Histological confirmation is not always feasible in patients with compromised respiratory status and/or coagulation abnormalities. Consequently, clinicians face a dilemma when it comes to the isolation of these fungi during patient follow-up: whether to treat or not to treat—this is the question. Due to the risk of mortality, therapeutic withholding is not an option. However, the available pharmacological options are limited, and preventive treatment exposes the patient to significant adverse effects and multiple drug interactions.

The objectives of this study were to analyze the frequency of isolation of *Scedosporium* spp. and *Lomentospora prolificans* in our lung transplant recipients, the clinical management of these isolations and the short- and long-term outcomes of patients. We also looked at the in vitro sensitivity of the isolates to the currently available antifungal drugs.

## 2. Materials and Methods

### 2.1. Study Design and Patient Population

We designed an observational, retrospective, and analytical study that included all patients aged ≥18 years transplanted at the Hospital Universitario Puerta de Hierro, Majadahonda (Madrid, Spain) between 1 January 2010 and 31 December 2023 who had post-surgical isolation of *Scedosporium* spp. or *Lomentospora prolificans*. The study protocol was approved by our center’s Ethics Committee on 27 January 2025 (No. 02/2025). Prior to the collection of data, patients were required to provide written informed consent.

In the pre-transplant period, sputum samples are periodically obtained for fungal and bacterial culture. On the day of the transplant, a bronchoscopy is routinely performed to obtain respiratory samples before the implantation of the graft. After the transplant, a bronchoscopy is performed routinely before the patient is discharged. Additional sputum samples and bronchoscopy are collected based on clinical evolution.

Patients are followed up after surgery in the outpatient clinic of the pulmonology department weekly for the first two weeks, monthly for the first three months, bimonthly for the first year and every three months thereafter.

Our standard approach to immunosuppression includes baliliximab and a life-time triple immunosuppression regimen with tacrolimus, mycopnenolate mofetil and prednisone.

Antibacterial prophylaxis is adjusted to the isolates obtained from samples of the donor and recipient at the time of surgery. The empirical administration of broad-spectrum antibiotics is suspended after 7–10 days.

Antifungal prophylaxis is based on the administration of nebulized liposomal amphotericin B (6 mL) every 48 h until discharge and weekly thereafter [11,12,13]. Our standard approach is to pre-emptively treat any fungal isolation. The antifungal combination initially chosen for patients with *Scedosporium* or *Lomentospora* isolation is voriconazole and terbinafine. Posaconazole or isavuconazole are chosen for patients intolerant to voriconazole or with relevant drug interactions.

### 2.2. Laboratory Fungal Isolation and Susceptibility Testing

Respiratory samples were cultivated on Sabouraud Chloramphenicol agar (Becton Dickinson™, Franklin Lakes, NJ, USA) and incubated at 25 °C and 37 °C for 14 days. Microorganism identification was performed by macroscopic and microscopic visualization as well as MALDI-TOF^®^ mass spectrometry (Bruker™, Berlin, Germany).

Species identification was confirmed at the Reference and Research Mycology Laboratory of the National Microbiology Center (ISCIII) by PCR amplification and sequencing of two regions (ITS region and β-tubulin). Susceptibility testing was performed by the microdilution method according to the EUCAST E. Def 9.4 protocol.

### 2.3. Definitions

The following definitions are based on the 2015 International Society for Heart and Lung Transplantation (ISHLT) Guidelines for the management of fungal infections in patients with mechanical circulatory support and cardiothoracic organ transplants. These definitions classify fungal presence and infections in transplant recipients as follows [6]:

Pre-transplant colonization: The presence of fungi in lower respiratory tract samples detected via culture prior to transplantation, in the absence of symptoms or radiological/endobronchial changes.

Post-transplant colonization: The presence of fungi in lower respiratory tract samples detected via culture after transplantation, in the absence of symptoms or radiological/endobronchial changes.

Localized infection: The presence of fungi in lower respiratory tract samples detected via culture, accompanied by symptoms and radiological or endobronchial changes.

Invasive fungal infection: The presence of fungi in lower respiratory tract samples detected via culture, accompanied by symptoms and radiological or endobronchial changes, along with histological evidence of tissue invasion.

Additionally, chronic lung allograft dysfunction (CLAD) is defined as a substantial and persistent decline (≥20%) in the measured forced expiratory volume (FEV_1_) value relative to the baseline. The baseline is determined as the mean of the two highest post-operative FEV_1_ measurements taken at least three weeks apart [14].

The electronic medical data and laboratory values recorded were demographic data, underlying disease, type of transplant, date of the first S/L isolate, antifungal treatment, diagnosis of CLAD and acute rejection and date and cause of death.

### 2.4. Statistical Analysis

Continuous variables are presented as mean (standard deviation, SD) or as median (interquartile range, IQR) when not normally distributed. Categorical variables are expressed as frequency (percentage, %). For univariable comparisons, the Student’s *t*-test or Mann–Whitney U test was used, depending on whether the assumptions of normality were met for numerical variables. For categorical data, Pearson’s chi-square test or Fisher’s exact test was applied, as appropriate. A 95% confidence interval (95% CI) was calculated to estimate the incidence of *Scedosporium* spp. or *Lomentospora prolificans* during the post-transplant follow-up period.

Post-transplant overall survival was analyzed using the Kaplan–Meier method, and group comparisons were assessed with the log-rank test. Statistical significance was defined as a *p*-value less than 0.05, and all tests were two-sided. Statistical analyses were performed using Stata version 18.0 (StataCorp. 2023. Stata Statistical Software: Release 18, StataCorp LLC, College Station, TX, USA).

## 3. Results

### 3.1. Characteristics of the Study Population

A total of 11 (2%, 95% CI 0.8; 3.0) of 576 patients who underwent lung transplantation in our unit between 1 January 2010 and 31 December 2023 had at least one isolation of *Scedosporium* spp. or *Lomentospora prolificans* (S/L) during the post-transplant follow-up period (Table 1).

The clinical and demographic characteristics of patients with or without S/L isolation are described in Table 2. No significant differences were observed between patients with and without isolates in relation to age, sex, underlying disease or type of transplant.

### 3.2. Microbiological Findings

During the study period, we obtained 143 S/L isolates. Samples with S/L isolation were as follows: bronchial aspirate (n = 65; 45%), sputum (n = 30; 21%), pleural fluid (n = 19; 13%), bronchoalveolar lavage (n = 14; 10%), blood (n = 4; 3%) and others (n = 11; 8%). The median time from transplantation to the first isolation was 535 days (IQR 157–1294 days). Three patients also had isolations of S/L prior to surgery.

The most frequently identified species were *L. prolificans* (four patients), *S. apiospermum* (three patients), *S. boydii* (three patients) and *S. ellipsoidea* (one patient). In two recipients, *S. apiospermum* and *L. prolificans* were isolated concomitantly in the same sample.

Coinfections with other microorganisms were detected in five recipients (45%), and two of them had more than one detected at the same time as S/L. The most frequent coinfected germens were Cytomegalovirus (n = 3; 27%), *Aspergillus* (n = 2; 18%), *Pseudomonas aeruginosa* (n = 1; 9%), *Stenotrophomonas maltophilia* (n = 1; 9%), *Staphylococcus aureus* (n = 1; 9%) and *Nocardia farcinica* (n = 1; 9%).

### 3.3. Type of Infection and Clinical Presentation

According to the ISHLT diagnostic criteria, the isolation was classified as colonization in four recipients (36%) and infection in seven (64%). The infection was limited to the lung in five patients and was disseminated in two. The fungus isolated most frequently in patients with pulmonary disease was the *S. apiospermum* complex, while *Lomentospora prolificans* was the fungus isolated in the two disseminated cases.

Radiological findings associated with the S/L infection were pulmonary infiltrates (n = 4; 57%), pleural effusion (n = 2; 29%), pulmonary nodules and/or micronodules (n = 2; 29%), mediastinitis (n = 1; 14%), tree-in-bud pattern (n = 1; 14%) and pulmonary thromboembolism (n = 1; 14%).

### 3.4. Antifungal Treatment and Outcomes

Ten of the eleven patients with isolation of S/L received antifungal treatment. Voriconazole (n = 8, 73%) and terbinafine (n = 6, 55%) were the most frequently used drugs. Other antifungals used were posaconazole (n = 3, 27%), echinocandins (n = 3, 27%) and amphotericin B (n = 2, 18%). Seven patients (70%) were treated with more than one drug, and the most common combination was voriconazole and terbinafine.

The eradication of fungi was achieved in seven patients (64%). Six of them received antifungal treatment and one experienced spontaneous resolution without treatment. Eradication was not achieved in two patients with pulmonary disease related to the *S. apiospermum* complex. Despite extensive combined antifungal treatment, two of the four patients with *L. prolificans* isolation died from disseminated disease.

During the post-transplant follow-up period, eight patients (73%) died. The cause of death was directly related to fungal infection in two recipients, both due to disseminated *Lomentospora prolificans* infection. Other causes of mortality included progressive chronic lung allograft dysfunction (four cases), COVID-19 (one case) and multiorgan failure (one case).

No significant differences in survival were observed among patients who had at least one post-transplant S/L isolate and those who did not (*p* log-rank test = 0.073) (Figure 1). However, the mortality rate among patients with *Lomentospora prolificans* infection was 50% for colonization and 100% for disseminated disease.

### 3.5. Antifungal Susceptibility Profiles

The in vitro antifungal susceptibility tests performed for *Lomentospora prolificans* and *Scedosporium* spp. against various antifungal agents are shown in Table 3.

*Lomentospora prolificans* showed in vitro resistance to all azoles, terbinafine and amphotericin B. The minimum inhibitory concentration (MIC) values for echinocandins ranged from 0.03 to >4 mg/L, with MIC_50_ values of 0.12 mg/L for anidulafungin, 1 mg/L for caspofungin and 0.06 mg/L for micafungin.

Regarding *Scedosporium apiospermum*, voriconazole exhibited the highest activity among azoles, with MIC values ranging from 0.25 to >8 mg/L and an MIC_50_ of 0.5 mg/L. The MIC_50_ values for amphotericin B and terbinafine were 4 mg/L and >16 mg/L, respectively. Regarding echinocandins, micafungin had an MIC_50_ of 0.06 mg/L, anidulafungin 0.12 mg/L and caspofungin 0.5 mg/L.

In relation to *Scedosporium boydii*, voriconazole was found to be the most active azole, with MIC values ranging from 0.5 to >8 mg/L and an MIC_50_ of 1.5 mg/L. In contrast, terbinafine demonstrated no in vitro antifungal activity, with MIC values > 16 mg/L. The MIC_50_ values for echinocandins were 1 mg/L for both caspofungin and anidulafungin and 0.25 mg/L for micafungin.

*Scedosporium ellipsoidea* exhibited MIC_50_ values of 0.5 mg/L for itraconazole, posaconazole and voriconazole, and 8 mg/L for isavuconazole. The MIC_50_ values for echinocandins were found to be 0.06 mg/L for anidulafungin, 0.25 mg/L for caspofungin and 0.03 mg/L for micafungin.

## 4. Discussion

In our study, we observed that the isolation of S/L is infrequent in lung transplant recipients. Nevertheless, its management remains very challenging due to the paucity of available therapeutic options. The majority of our patients developed localized pulmonary infections with positive outcomes. However, two of the four recipients infected with *L. prolificans* died of disseminated disease despite the administration of several combined antifungal drugs. The in vitro susceptibility tests performed showed resistance to the majority of antifungal drugs currently available.

Lung transplant recipients have an increased susceptibility to lower respiratory tract infections with S/L due to the continuous contact of the graft with air and the profound immunosuppression required to prevent acute and chronic rejection [15]. Moreover, the development of these multi-drug fungal infections is also promoted by the frequent exposure of these patients to broad-spectrum antifungal drugs during prophylaxis or empirical treatments. The percentage of isolation of S/L observed in our study was low (2%) and similar to that described by other authors [16,17]. The diagnostic value of these isolates should always be interpreted with caution in conjunction with clinical, radiological and histological findings. However, since invasive infection has been associated with a high mortality rate, the protocol in our center is to treat all patients with any fungal isolation. That is the reason why antifungal treatment was started for each patient except for one who eradicated the germ spontaneously.

Differentiation between colonization and infection can be complex. After applying the ISHLT diagnostic criteria [6], the isolates of S/L were classified as colonization in 36% of our patients and infection in 64%. The five cases classified as localized pulmonary disease were caused by *S. apiospermum* complex (*S. apiospermum* and *S. boydii*) and successfully treated with antifungal drugs. However, despite several combined antifungal treatments, two of the four patients infected with *L. prolificans* died as a result of hematogenous dissemination of the disease.

The capacity of *L. prolificans* to germinate, form conidia, penetrate blood vessels and disseminate rapidly in host tissues could provide a plausible explanation for the rapid progression of the infection and its high mortality rate. Furthermore, the intrinsic resistance of the fungi to most of the antifungal drugs currently available limits therapeutic options.

Four patients in our study had isolates of *L. prolificans*, three before and one after surgery. The isolation in the pre-transplant period is considered a poor prognostic factor, and there is controversy regarding the contraindication of transplantation in previously colonized patients [18,19]. Two patients with cystic fibrosis had *L. prolificans* isolation prior to surgery and were transplanted under preventive treatment with voriconazole. One of them eradicated the fungi after surgery, but the other died due to disseminated disease regardless of the antifungal treatment.

The isolation of *L. prolificans* in the third case was achieved from samples obtained in the operating room before the implantation of the graft. The patient underwent an emergency transplant due to pulmonary fibrosis and died in the post-transplant period despite the administration of several antifungal drugs.

The fourth patient with *L. prolificans* acquired the infection in the post-transplant period and was able to achieve eradication with a combined course of treatment.

The mortality rate associated with *L. prolificans* isolation in our patients was 50% for colonization and 100% for disseminated disease, which is consistent with the findings observed by other authors [20,21,22]. In order to reduce mortality and avoid unnecessary treatments, it is essential to improve the currently available diagnostic criteria for disease [23,24,25].

Regarding the susceptibility profiles, our results are similar to those previously described [26,27]. The *S. apiospermum* complex exhibits resistance to amphotericin B and reduced susceptibility to echinocandins [8]. Voriconazole was found to be the most active azole, with MIC_50_ values of 0.5 mg/L for *S. apiospermum* and 1.5 mg/L for *S. boydii*. These values were lower than those of isavuconazole and posaconazole, a finding that has been previously described by other authors. In this regard, our results support the use of voriconazole as a first-line antifungal for the treatment of infections caused by these fungi [23,24,26,28]. Most of our patients received combination therapy. While the synergy and fungicidal effect of the combination of terbinafine and azoles has been demonstrated in in vitro studies, no greater efficacy has been shown in vivo when these drugs are combined with other antifungals, such as terbinafine or echinocandins [15,23,25,29].

The first-line treatment recommended by the guidelines for patients with *L. prolificans* infection is the combination of voriconazole and terbinafine [23,24,25]. This combination has demonstrated superior response rates and better survival outcomes in in vitro antifungal synergy [30]. In the present study, the strains of *L. prolificans* isolated exhibited panfungal resistance to azoles, terbinafine and amphotericin B, while echinocandins demonstrated moderate in vitro activity, which is consistent with previously reported findings [26].

In addition to exhibiting reduced sensitivity to antifungal agents commonly used in clinical practice, these fungi are influenced by several factors that contribute to resistance and therapeutic failure, including host immunosuppression, lack of source control or suboptimal antifungal pharmacokinetics [31]. The high mortality rate of these infections highlights the need for the development of new antifungal drugs and the promotion of multicenter clinical trials with drugs such as fosmanogepix and olorofim, which have shown good in vitro activity against *Scedosporium* and *Lomentospora* [32,33,34]. Notably, in vitro studies evaluating the efficacy of olorofim against *Scedosporium*/*Lomentospora* have reported MICs lower than those observed for voriconazole and posaconazole [15,35]. In addition to their in vitro efficacy, these agents have exhibited a positive response in patients with severe disseminated infections, thus supporting their potential for use in clinical practice [36,37]. However, the low number of infections produced by these fungi limits the development of clinical trials [38].

### Limitations

The major limitation of this study was its retrospective and single-center nature. The low number of *Scedosporium*/*Lomentospora* isolations did not allow us to identify risk factors. Comparison to other studies was difficult due to the complexity of differentiating between colonization and infection. Our findings are a retrospective experience of a clinical practice, and the great diversity of antifungal drugs and combinations used throughout the study made it very difficult to draw conclusions in relation to efficacy.

Fungal cultures were performed using Sabouraud Chloramphenicol agar. Although specific media can enhance *Scedosporium*/*Lomentospora* isolation by inhibiting faster-growing fungi like *Aspergillus*, especially in cystic fibrosis patients [39,40], our detection rates were consistent with previous studies [16,17].

## 5. Conclusions

Infections with *Scedosporium* spp. and *Lomentospora prolificans* are infrequent. The high mortality due to invasive disease highlights the need for early diagnosis and appropriate antifungal treatment. To improve the prognosis of these infections it is necessary to develop new diagnostic and therapeutic tools.

## Figures and Tables

**Figure 1 jof-11-00291-f001:**
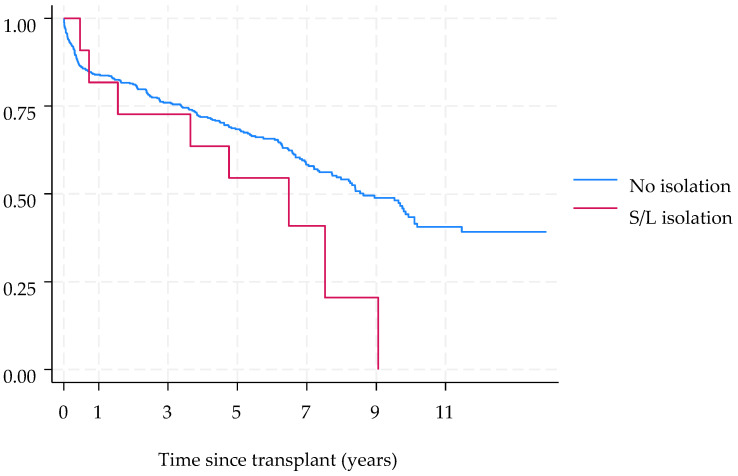
Survival of lung transplant recipients according to *Scedosporium* spp. and *L. prolificans* isolation.

**Table 1 jof-11-00291-t001:** *Scedosporium* spp. and *Lomentospora prolificans* in lung transplant recipients: clinical features, microbiological findings and outcomes.

	Type of Transplant	Age	Gender	Underlying Disease	Species	Time since LTx	Infection/ Colonization	Eradication	Sample (n)	Other Microorganisms	Treatment	Radiological Findings	Outcome	Cause Related to FI
1	Bilateral Retransplantation	40	M	Cystic Fibrosis	*Scedosporium boydii*	1 y, 5 m	Post-transplant colonization	Yes	BAS (2)	No	No	None	Alive	
2	Bilateral	57	M	COPD	*Scedosporium* *apiospermum*	2 y, 1 m	Localized infection	No	Sputum (8) BAS (7) BAL (3)	No	VOR (IV)	Pulmonary infiltrates	Deceased	No
3	Bilateral	67	M	ILD	*Lomentospora prolificans*	1 y, 2 m	Localized infection	Yes	Sputum (1) BAS (5) BAL (3)	*Aspergillus* *Terreus* *Nocardia farcinica* CMV	VOR (PO, INH) TER (PO)	Nodular changes	Deceased	No
4	Bilateral	72	F	ILD	*Scedosporium boydii*	6 y, 9 m	Localized infection	Yes	Sputum (2) BAS (3) BAL (2)	No	ISA (IV) AMB (INH) OLO (PO)	Pulmonary infiltrates	Deceased	No
5	Bilateral	55	M	ILD	*Lomentospora prolificans*	0 d	Disseminated infection (pre-transplant colonization and post-transplant infection)	No	Sputum (3) BAS (10) BAL (4) Pleural fluid (7) Bronchial biopsy (2) Blood (4)	No	VOR (PO, INH, IP) TER (PO) ANI (IV) AMB (IP) OLO (PO)	Pleural effusion and micronodules	Deceased	Yes
6	Bilateral	25	M	Cystic Fibrosis	*Lomentospora* *prolificans*, *Scedosporium* *apiospermum* and *S. boydii*	0 d	Disseminated infection (pre-transplant colonization and post-transplant infection)	No	Sputum (3) BAS (36) BAL (1) Pleural fluid (14) Pleural biopsy (1) Catheter (1) Wound (2)	*Pseudomonas* *aeruginosa*	VOR (PO, IV) POS (PO) ANI (IV) TER (PO)	Pulmonary infiltrates, pleural effusion, mediastinitis, pulmonary thromboembolism	Deceased	Yes
7	Bilateral	64	F	ILD	*Scedosporium* *ellipsoidea*	3 y, 1 m	Post-transplant colonization	Yes	BAL (1)	*Staphylococcus* *aureus*	VOR (INH)	None	Alive	
8	Bilateral	17	F	Cystic Fibrosis	*Lomentospora prolificans* and *Scedosporium* *apiospermum*	7 d	Pre- and post-transplant colonization	Yes	Sputum (7) BAS (2)	CMV	POS (IV) VOR (INH) TER (PO)	None	Alive	
9	Unilateral	70	M	ILD	*Scedosporium* *apiospermum*	3 y, 11 m	Post-transplant colonization	Yes	BAS (1)	No	VOR (INH) TER (PO)	None	Deceased	No
10	Bilateral	63	M	ILD	*Scedosporium* *apiospermum*	9 m	Localized infection	Yes	Sputum (2) Bronchial biopsy (1) BAS (1)	*Stenotrophomonas maltophilia*, CMV	VOR (IV)	Pulmonary infiltrates	Deceased	No
11	Bilateral	64	M	Bronchiectasis	*Scedosporium boydii*	6 y, 4 m	Localized infection	No	Sputum (4)	*Aspergillus terreus*	POS (IV) TER (PO) MIC (IV)	Tree-in-bud pattern	Deceased	No

Abbreviations: M (Male); F (Female); COPD (Chronic Obstructive Pulmonary Disease); ILD (Diffuse Interstitial Lung Disease); BAS (bronchial aspirate); BAL (bronchial lavage); CMV (Cytomegalovirus); VOR (voriconazole); TER (terbinafine); ISA (isavuconazole); AMB (amphotericin b); POS (posaconazole); ANI (anidulafungin); MIC (micafungin); OLO (olorofim); y (years); m (months); FI (Fungal Infection); IV (intravenous); INH (inhalation); PO (oral).

**Table 2 jof-11-00291-t002:** Demographic characteristics according to S/L isolation.

	S/L Isolation			
	No	Yes	Total	
**Total (n, %)**	565 (98%)	11 (2%)	576 (100%)	
**Age (mean ± SD)**	56.09 ± 10.31	52 ± 16.03	56.51 ± 10.43	*p* = 0.198
**Gender (n, %)**				
Male	365 (65%)	8 (73%)	373 (65%)	
Female	200 (35%)	3 (27%)	203 (35%)	
**Underlying disease (n, %)**				
COPD/emphysema	227 (40%)	1 (9%)	228 (40%)	
ILD	253 (45%)	6 (55%)	259 (44%)	
Cystic fibrosis	41 (7%)	3 (27%)	44 (8%)	
Others	44 (8%)	1 (9%)	45 (8%)	
**Type of transplant (n, %)**				
Bilateral lung	487 (86%)	10 (91%)	497 (91%)	
Unilateral lung	78 (14%)	1 (9%)	79 (9%)	

Abbreviations: COPD (Chronic Obstructive Pulmonary Disease); ILD (Diffuse Interstitial Lung Disease).

**Table 3 jof-11-00291-t003:** In vitro susceptibilities of *Scedosporium apiospermum* isolates, *Scedosporium boydii* isolates and *Lomentospora prolificans* isolates.

	*S. apiospermum* (mg/L)			*S. boydii* (mg/L)			*L. prolificans* (mg/L)		
	No. Tested	Range	MIC_50_	No. Tested	Range	MIC_50_	No. Tested	Range	MIC_50_
Voriconazole	9	0.25–>8	0.5	6	0.5–>8	1.5	6	>8	>8
Isavuconazole	3	8–>8	8	5	2–>8	>8	3	>8	>8
Itraconazole	9	8–>8	>8	6	>8	>8	6	>8	>8
Posaconazole	9	0.25–>8	1	6	0.5–>8	>8	6	>8	>8
Amphotericin B	9	4–>16	4	6	2–>16	>16	6	8–>16	>16
Terbinafine	8	16–>16	>16	6	>16	>16	6	>16	>16
Anidulafungin	4	0.015–1	0.12	6	0.25–4	0,25	5	0.03–>4	0.12
Caspofungin	9	0.5–>16	0.5	6	0.5–8	1	6	0.12–>16	1
Micafungin	4	0.03–0.25	0.06	6	0.12–0.5	0.25	5	0.03–>2	0.06

The “No. tested” column indicates the number of isolates tested for each antifungal agent. In some patients with more than one isolate, several isolates were tested independently.

## Data Availability

The original contributions presented in this study are included in the article. Further inquiries can be directed to the corresponding author.

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
