# Peer review of "Clinical Significance and Therapeutic Challenges of Scedosporium spp. and Lomentospora prolificans Isolates in a Single-Center Cohort of Lung Transplant Recipients"

_jof, 2025, doi:10.3390/jof11040291_

Round 1
Reviewer 1 Report
This is a useful study that is presented clearly. As described in the limitations section, the low numbers of patients with these rare but important infections in this centre prevents more being made of the findings.
lines 313-314:
“as fosmanogepix and olorofim, which have shown good in vitro activity against Scedosporium and Lomentospora [17–19]. However, the low number of infections produced by these fungi limits the development of clinical trials”
References should be 26-28 and not 17-19.
These new agents have also had some clinical cases published that that the authors should consider referencing eg PMID: 39265435 and PMID: 39471308.
Author Response
Thank you very much for taking the time to review our manuscript and for providing your valuable feedback. We truly appreciate your insightful comments and suggestions, which have greatly contributed to improving the quality of our work.
Please find our detailed responses below, with corresponding revisions highlighted in the re-submitted files.
lines 313-314:
“as fosmanogepix and olorofim, which have shown good in vitro activity against Scedosporium and Lomentospora [17–19]. However, the low number of infections produced by these fungi limits the development of clinical trials”
References should be 26-28 and not 17-19.
These new agents have also had some clinical cases published that that the authors should consider referencing eg PMID: 39265435 and PMID: 39471308.
We have corrected the citation error. The references you suggested were highly valuable, and we greatly appreciated your recommendation to include them. These have now been incorporated into paragraph 4, page 12, lines 326–329.
Thank you again for your guidance and for facilitating the improvement of our manuscript. We look forward to any further comments you may have.
Reviewer 2 Report
The objectives addressed in the paper are of great interest to the medical community, as S/L infections are an emerging problem and are challenging to treat. However, I would suggest addressing the following points/comments (please, see Detail comments)
1) An interesting point would be to comment on whether there were differences in the number of cases in the pre-COVID vs post-COVID era, or if there was an increase in the number of cases in recent years, or if the number remained relatively the same over the years, given that the study period ranged from 2010 to 2023.
2) The authors should include data to justify why they use nebulized ambisome prophylaxis in these patients.
3)If authors aim to assess the prevalence of Scedosporium, using only Sabouraud agar as a culture medium is not appropriate. There are more selective media, such as Scedosporium agar (Sce-Sel-agar), and this information should be included in the section where they discuss the limitations of the study.
4) Why do they incubate the samples for only two weeks instead of four weeks?
5) Did they measure galactomannan (GM) in these patients?
6)In line 107, please specify what type of samples were cultured (respiratory?).
7) In line 163, they mention obtaining 143 isolates; from what? Fungi? Please be more specific.
8) For the patient who did not receive treatment, how many cultures tested positive for Scedosporium? Were they contaminants? I find it striking that the infection resolved without treatment. If this is the case, please look for relevant literature on this in immunosuppressed patients and included in the manuscript.
Author Response
We sincerely appreciate your valuable feedback and insightful suggestions, which have helped us improve the clarity and depth of our manuscript. Below, we address each of your comments:
- Title Modification Thank you for your suggestion regarding the title. We agree that the term "large cohort" may be misleading in the context of a single-center study. We have modified the title accordingly to: Clinical significance and therapeutic challenges of Scedosporium spp. and Lomentospora prolificans isolates in a single-centre cohort of lung transplant recipients in Spain.
- Pre- vs. Post-COVID Era Case Distribution We appreciate your suggestion to analyze the temporal trends in case distribution. In our study, we observed that regarding the differences in S/L isolation before and after the SARS-CoV-2 pandemic, S/L was isolated in six patients (55%) during the pre-pandemic period and in five patients (45%) after the onset of the COVID-19 pandemic.
- Justification for Nebulized Ambisome Prophylaxis Thank you for your suggestion. We have included references supporting the safety and efficacy of antifungal prophylaxis with Ambisome post-surgery in the Materials and Methods section (paragraph 4, page 3, lines 99–100).
- Use of Selective Culture Media Thank you for your comment regarding the culture medium. We acknowledge that selective media such as Sce-Sel-agar improve the sensitivity for detecting Scedosporium spp. and Lomentospora prolificans, particularly in patients with cystic fibrosis. However, despite using Sabouraud Chloramphenicol agar, our detection rates align with those reported in other studies. We have now included this point in the limitations section, citing relevant literature.
- Incubation Duration In our laboratory, we incubate fungal cultures for two weeks for patients without risk factors for endemic mycoses. However, for slow-growing and thermally dimorphic fungi (e.g., Histoplasma spp., Paracoccidioides spp., Blastomyces dermatitidis), we extend incubation up to 40 days.
- Galactomannan Testing Galactomannan was only performed in three patients, all of whom tested negative. Additionally, one patient had a negative β-D-glucan test. We acknowledge the limitations of serum GM testing in non-neutropenic individuals and we don’t performed it routinaly.
- Clarification of Sample Type (Line 107) We have specified that the samples cultured were respiratory specimens.
- Clarification of Isolates (Line 163) The 143 isolates correspond specifically to Scedosporium spp. and Lomentospora prolificans, and we have now clarified this in the text.
- Untreated Patient and Literature Support The patient who did not receive treatment was considered to have colonization rather than infection. Subsequent cultures from this patient did not yield Scedosporium spp., and antifungal therapy was not deemed necessary.
Thank you again for your guidance and for helping to strengthen the manuscript. We look forward to any additional comments you may have.
Reviewer 3 Report
The work addresses a critical issue in lung transplant recipients, highlighting the high mortality of Lomentospora prolificans infections. The authors provide valuable data on fungal isolation frequency, clinical outcomes, and antifungal resistance and emphasizes the need for new antifungal therapies.
- In order to expand the sample size authors should consider a multicenter study for better representativeness.
- Compare outcomes with patients without fungal isolation (control group).
- Improve statistical analysis using regression models and propensity score matching.
- I recommend to include therapeutic drug monitoring for voriconazole.
- Clarify infection classification using additional biomarkers like β-D-glucan.
- Strengthen the discussion expanding on antifungal resistance implications and alternative treatments.
Author Response
Thank you very much for your time and thoughtful review of our manuscript. We are grateful for your insightful and constructive comments and have been highly valuable in enhancing the quality of our work.
Please find our detailed responses below, with corresponding revisions highlighted in the re-submitted files.
- Authors should consider a multicenter study to expand sample size and improve generalizability.
Response: Thank you for this excellent suggestion. We completely agree that conducting a multicenter study would significantly enrich the data and enhance representativeness. We believe it would be an exciting avenue for future research.
2) Comparison with a control group (patients without fungal isolation) is recommended.
Response: Thank you for this important point. Notably, overall survival was similar between groups.
3) Strengthen the statistical analysis by incorporating regression models or propensity score matching.
Response: We truly appreciate your input regarding the statistical approach. As you rightly know, statistical methods must be tailored to the dataset's characteristics. Given that only 11 patients in our cohort presented with Scedosporium/Lomentospora isolation, applying regression models or propensity score analysis would not be appropriate or statistically reliable. We recognize the limitations imposed by our sample size, and for this reason, we opted for a descriptive and univariate analysis, which we believe is the most suitable and rigorous approach under the circumstances. We have now clarified this in the revised Statistical Analysis section.
4) I recommend including data on TDM for voriconazole.
Response: Thank you for this helpful suggestion. Unfortunately, voriconazole levels were not routinely measured in our patients.
5) Consider using additional biomarkers to classify fungal infections.
Response: Thank you for raising this point. In our clinical setting, we did not routinely use fungal biomarkers, such as β-D-glucan, for diagnosing invasive fungal infections in lung transplant recipients.
6) Strengthen the discussion by elaborating on antifungal resistance and available therapeutic alternatives.
Response: Thank you once again for this valuable recommendation. We have expanded the discussion on this topic in page 11, lines 308–316, to address the clinical implications of antifungal resistance and discuss emerging therapies.
Thank you again for your insightful feedback and for helping us to improve our manuscript. Your comments have enriched the final version, and we hope it meets your expectations. We remain at your disposal for any further clarifications or suggestions.
Round 2
Reviewer 2 Report
Dear authors,
Thank you very much for your detailed answers to all my requested issues. I consider the manuscript has been sufficiently improved to warrant publication in JoF.
I have no more comments for the revised version of the manuscript.
Author Response
We sincerely thank you for your insightful comments and constructive suggestions, which have significantly enhanced the quality of our manuscript.
Reviewer 3 Report
The authors have addressed all the suggested changes.
The authors have addressed all the suggested changes.
Author Response

(The authors gave the same response as above.)
